# Operational Feasibility of Point-of-Care Testing for Sickle Cell Disease in Resource-Limited Settings of Tribal Sub-Plan Region of India

**DOI:** 10.3390/diagnostics15030348

**Published:** 2025-02-02

**Authors:** Mahendra Thakor, Janesh Kumar Gautam, Ansuman Panigrahi, Dharmendra Garasiya, Shankar Lal Brhamnia, Suman Sundar Mohanty

**Affiliations:** 1ICMR-National Institute for Implementation Research on Non-Communicable Diseases, New Pali Road, Jodhpur 342005, India; mahendra15519@gmail.com (M.T.); janeshgautam@gmail.com (J.K.G.); dr.ansuman3@gmail.com (A.P.); 2Office of Block Chief Medical Officer, Jhadol, Udaipur 313702, India bcmo_jhadol@yahoo.com; 3Office of Block Chief Medical Officer, Kotda, Udaipur 385535, India; bcmo_kotra@yahoo.com

**Keywords:** competence, diagnosis, healthcare worker, point-of-care, sickle cell disease

## Abstract

**Background**: Sickle cell disease (SCD) individuals in India are mostly identified when they become symptomatic. To provide a timely diagnosis of SCD to participants, healthcare workers should be competent in using the point-of-care test (POCT). In this study, we aimed to evaluate the competence of healthcare workers to screen infants and adult populations through POCT. **Methodology**: This study was conducted in pilot mode over 8 months from April to November 2023. A random sampling method was used to select ten auxiliary nursing midwives (ANMs), ten lab technicians (LTs), and five medical officers (MOs). Each selected ANM and LT was supposed to conduct ten tests and MOs to conduct five tests. The POCT used to diagnose sickle cell disease was HemoTypeSC. **Results**: Among the healthcare workers who participated in the study, 67% belonged to the scheduled tribes. When the ANM and LT competencies were compared for the pre-analytical phase (phase I), ANMs were more competent than the LTs. ANMs were more adept at handling people, whereas the LTs were more competent in conducting the test procedures. When the comparison was made for the analytical phase (phase II), both the ANMs and LTs were found to be equally competent. ANMs followed the standard operating procedure (SOP) more precisely than MOs and LTs. In the post-analytical phase, LTs were found to be more competent than ANMs. The approach used in this study with sub-centers and primary health centers (PHCs) appears to have encouraged the feasibility of the screening program. **Conclusions**: The results of this study conclude that the healthcare workers in the region are competent to perform the POCT for the diagnosis of sickle cell disease. The POCT may be introduced in the program for the diagnosis of SCD.

## 1. Introduction

Sickle cell disease (SCD) is the most prevalent and common inherited blood disorder [1]. In developed countries, widespread screening and early intervention for SCD have been shown to decrease childhood mortality drastically [2]. However, in developing countries and low-resource settings, SCD goes undiagnosed until the patients exhibit clinical signs and symptoms [3]. Additionally, it is difficult to diagnose disease in affected patients in an urgent care setting in both high- and low-resource settings, which frequently causes treatment to be delayed. This leads to a delay in the treatment of SCD patients. All current diagnostic techniques are expensive and time-consuming and require skilled technicians and high-end infrastructure. To fill this gap, point-of-care tests (POCT) have been developed. POCT kits have transformed medical care for a variety of diseases, from infectious diseases like malaria to genetic disorders like SCD, by offering timely, actionable results for prompt management. However, in developing nations, where the need is greatest, they are underutilized in diagnosing SCD [4].

Since there is no national neonatal screening program for sickle cell disease in India, the affected children are typically discovered when they start exhibiting symptoms. However, in some areas, a few newborn screening programs have been started in the past five to six years [5,6,7,8,9]. The lack of simple and accessible SCD diagnostics is most likely the cause of the postponement of a neonatal screening program. A POCT with high specificity and sensitivity that is affordable, dependable, and simple to use is required. Among the many benefits would be (i) ease of use by medical personnel in low-resource environments; (ii) quick delivery of results, allowing for timely patient notification and, if required, family counseling; and (iii) the ability to use the test in remote locations, resulting in early diagnosis and a decrease in mortality and morbidity through the use of therapeutic interventions, particularly antimalarials and penicillin prophylaxis. Although a number of POCTs for SCD have recently been created and verified, HemoTypeSC, one of the most well-known tests, has undergone extensive testing [10].

It has been demonstrated that HemoTypeSC is 100% accurate in determining the correct Hb phenotype [11]. These antibodies can accurately identify newborns with elevated HbF but low levels of HbA or HbS because they are blind to hemoglobin F (HbF). Nnodu et al. [10] conducted a study in Nigeria to assess the accuracy, specificity, and sensitivity of HemoTypeSC in detecting Hb phenotypes (AA, AS, AC, SS, SC, and CC) at various primary healthcare centers in a real-world field setting. They used the gold standard, high-performance liquid chromatography (HPLC), and found that the sensitivity and specificity for HbS and HbC were 100%. Therefore, the POCT was selected for this study. Health departments across many Indian states are considering the POCT (lateral flow device) for diagnosing SCD due to its advantages [12]. It is affordable and cheap (each kit costs less than USD 0.5), the test kit does not require much training or specific skills, it is user-friendly, and it requires only a small blood sample (1–5 μL) from a finger prick. The POCT delivers results within 10 min, offering a quicker, simpler alternative to HPLC, which requires more blood and complex equipment [10]. Additionally, it is portable and does not need electricity, making it suitable for use in various settings.

We aimed to assess the competence of healthcare workers to use the POCTs currently available in primary healthcare centers for routine diagnosis and follow-up of infants with sickle cell disease. This study will assess the competence of healthcare workers in testing infants, adults, and pregnant women for sickle cell disorder using point-of-care devices.

## 2. Methodology

Study Design: This study was cross-sectional in nature and was conducted to assess the competence of healthcare workers for the diagnosis of SCD through HemotypeSC.

Study Duration: This study was conducted over 8 months from April to November 2023.

Study Population: The healthcare workers included in this study were auxiliary nurse midwives (ANM), laboratory technicians (LT), and medical officers (MO) of the Jhadol and Kotda Blocks of the Udaipur District of Rajasthan.

Sample Size: A random sampling method was used to select ANMs, LTs, and MOs. Ten ANMs, ten LTs, and five MOs were selected for the study. Each selected ANM and LT was supposed to conduct ten POC tests. Each of the selected MOs was supposed to conduct five POC tests.

Study site: This study was conducted in Rajasthan, India’s tribal sub-plan area. The tribal sub-plan region was chosen based on three criteria: the size, compactness, and proportion of the tribal people to the overall population (more than 50%). Pratapgarh, Banswara, Dungarpur, Udaipur (partial), and the Abu Road blocks of the Sirohi District are all included in Rajasthan’s scheduled area [1]. The Kotra and Jhadol tehsils of the Udaipur District were the study sites (Figure 1). A tehsil is a sub-division of a district that is responsible for the administration and revenue collection of a particular area within the district, an important part of the local governance structure, and plays a crucial role in the development and administration of its local community. The Kotra tehsil consists of 262 revenue villages and 31 panchayats. The Kotra tehsil of the Udaipur District has a total population of 230,532 as per the 2011 census. The Jhadol tehsil consists of 283 revenue villages and 45 panchayats. The Jhadol tehsil has a population of 249,297 people.

HemotypeSC: The HemoTypeSC test kits, which included a volumetric inoculation loop, a sample cup, a transfer dropper, and the lateral flow assay (LFA) test strip, were kept at room temperature. The study location typically experiences a temperature of 24–32 °C and 62–81% humidity during the study months. HemoTypeSC does not need to be refrigerated and is thought to be stable at high temperatures.

Procedure: Advocacy of the training program for healthcare workers for the SCD diagnosis was conducted at the district level. An official letter was sent to the Chief Medical Health Officer to depute the healthcare workers to receive sickle cell diagnosis training. A meeting with the Chief Medical Health Officer of the district was held for the finalization of the venue and the time of training. The agenda was finalized in consultation with the Block’s Chief Medical Health Officer. The study’s selected ANMs, LTs, and MOs were invited for training through an official letter. The detailed protocol for the training of the healthcare workers is given below.

Healthcare workers, including ANMs, LTs, and MOs, were trained on the HemoTypeSC™ rapid test kit during a half-day workshop, which focused on identifying crucial hemoglobin variants such as HbAA, HbSS, HbSC, HbCC, HbAS, and HbAC for screening sickle cell disease and other hemoglobinopathies. The training covered the kit’s components—test strips, blood sampling devices, and dropper pipettes—and emphasized biohazard safety in the setup of the testing area. It was noted that additional infrastructure at the healthcare facilities like potable water, timers, lancing devices, test vials, and racks was necessary but did not exist and was thus provided by the research team.

The POCT testing platform was established at PHC/sub-center locations, where a step-by-step approach for using the HemoTypeSC was demonstrated. During practical sessions, trainees practiced blood sample collection, ensuring that the white pad of the blood sampling device was fully saturated. They learned the procedure steps: adding water to the test vial, obtaining a blood sample, swirling the sample in the vial to mix, and then inserting and reading the test strip against a results chart after 10 min.

Before conducting the SCD testing, consent was obtained from all eligible participants. The results of the HemoTypeSC were then compared with those from HPLC testing. Participants who tested positive were subsequently referred to the nearby PHC. The training emphasized correct technique, result interpretation, and post-test procedures, including the disposal of used components, all to ensure proficiency in using the HemoTypeSC™ test kit effectively.

Diagnosis of SCD: The point-of-care test (HemoTypeSC, Silver Lake Research, Azusa, CA, USA) was used to identify SCD in all infants, children, and adults who visited the sub-centers and primary health centers of the study sites. Blood samples from eight-week-old infants were drawn by heel-prick, while for others, finger-prick collection was used. For testing, the HemoTypeSC absorbent pad absorbed around 1 μL of blood. The process was carried out according to the manufacturer’s instructions. Additional blood was taken by capillary into labeled tubes for the confirmation of testing of samples through HPLC. HPLC of whole blood samples of positive cases and negative cases was carried out using Cation Exchange HPLC (CE-HPLC). CE-HPLC was performed on BioRad variant II (Bio-Rad Laboratories, Inc., Hercules, CA, USA) using β-thal short program. Commercially available controls were used to calibrate the instrument, and samples were periodically cross-checked for quality assurance. The areas and time of retention of chromatography were examined for the identification of different Hb variants.

The POCT test results were matched with HPLC results for the competence testing of HCW. All the trait and disease samples were run through the HPLC and 10% of the normal samples were also run through the HPLC. The results of the test were provided to the participants and parents of the children by the ANMs. Post-screening counseling was provided to them, and they were referred to a doctor. The diagnosis of SCD in pregnant women was performed during antenatal checkup (ANC) by MOs, LTs, and ANMs. Each ANM conducted the test on 10 participants. Each LT carried out SCD POCT on 10 different participants to diagnose sickle cell disease. Similarly, the medical officer conducted SCD POCT on five different participants to diagnose sickle cell disease.

Diagnosis of SCD of pregnant women during antenatal checkup (ANC): As per the national guidelines, a pregnant woman has to make four visits for antenatal checks during pregnancy. The first visit is recommended as soon as the pregnancy is suspected. The second, third, and fourth visits should be scheduled around 26, 32, and 36 weeks, respectively [13]. ANC facilities in rural areas are provided through the health sub-center. As previously stated, the point-of-care HemoTypeSC (Silver Lake Research, Azusa, CA, USA) test was used to screen them for sickle cell disease. The POCT findings were matched with HPLC results for competence testing. The results of the diagnosis test were given to the participants by the technician. Doctors conducted post-screening counseling. Each LT carried out the point-of-care test on 10 different participants to diagnose sickle cell disease. Similarly, medical officers conducted a point-of-care test on five different participants to diagnose sickle cell disease. An additional 0.5 mL of blood was used for the confirmation of the test.

Competence testing/Accuracy: An observation checklist was prepared and tested to record the competence of the healthcare workers for the diagnosis of SCD through the POCT. The checklist was prepared after group discussion and pre-testing of the observational points. The checklist was composed of 15 observation points. The observation checklist was divided into patients’ safety (pre-analytical phase), an analytical phase, and a post-analytical phase. Each part has five checkpoints and the credit of one mark for each checklist point in the pre-analytical phase, two marks for the analytical phase, and three marks for the post-analytical phase. During the performance of the POCT, observation was recorded by the senior investigator, and marks were given. A minimum of 60% marking was kept as the qualifier for conducting POCT in the future. A comparison of the result was performed with the HPLC method for the competence testing.

## 3. Assessment

The researchers in the team computed HemoTypeSC’s sensitivity, specificity, positive and negative predictive values, and overall accuracy compared to the “gold standard” (HPLC). Overall accuracy was defined as (prevalence×sensitivity)/(1 prevalence)(specificity), where TP = number of true positive events, FP = number of false positive events, and TN = number of true negative events. Sensitivity was defined as TP/(FN + TP) × 100, specificity as TN/(FP + TN) × 100, positive predictive value as TP/(TP + FP) × 100, and negative predictive value as TN/(TN + FN) × 100.

## 4. Results

Study Population: The study population consisted of 25 healthcare workers. Among the 25 healthcare workers, ten, ten, and five belonged to ANMs, LTs, and MOs, respectively (Table 1). Most of the healthcare workers (68%) belonged to the scheduled tribe community (Table 1). The demographic details of the healthcare workers (sex, category, age group, education qualifications, and service rendered) are given in Table 1. Their mean (±SD) age was 32.03 ± 7.5 years. Most of the participants (52%) were 18–28 years (Table 1). All the ANMs and laboratory technicians had diploma degrees in nursing and medical technology, respectively. Medical officers had Bachelor of Medicine and Bachelor of Surgery (MBBS) degrees.

When the competence of the healthcare workers was compared for the pre-analytical phase (phase I), ANMs and LTs were found to be less competent than the MOs (Table 2). ANMs were more adept in wearing their personal protection equipment. The LTs were more competent in disinfecting the working surfaces and body parts of the participants. When the comparison was made for the analytical phase (phase II), both the MOs and LTs were found to be equally competent and better than ANMs. ANMs were not as comfortable handling the kit as MOs and LTs. The MOs properly conducted the procedure for conducting the test with the kit. In the post-analytical phase, LTs were found to be more competent than ANMs in the interpretation of the results and maintaining the cleanliness of the work area. Medical officers provided counseling to the parents of affected SCD infants. LTs and MOs managed the biomedical waste and decontaminated surfaces better than ANMs. However, ANMs interpreted and noted the results more accurately after conducting a few tests. ANMs counseled the affected infants better than the LTs. When the overall competence of the healthcare workers was compared (Table 2), medical officers were found to be more competent than the ANMs and LTs in all three categories (patients’ safety (pre-analytical phase), analytical phase). The analysis in Table 2 revealed significant differences in adherence to healthcare practices across the pre-analytical, analytical, and post-analytical phases among LTs, MOs, and ANMs. LTs and MOs consistently showed higher compliance compared to ANMs in several key areas, such as managing biomedical waste and decontaminating surfaces. The analytical phase showed fewer differences in the pre- and post-analytical phases, which highlighted areas requiring targeted interventions to ensure consistent adherence to standard practices, particularly among ANM workers.

A total of 238 participants participated on a first-come basis for the diagnosis of SCD. Among the 238 participants, 107 (44.9%) were children, 37 (15.5%) adolescents, and 94 (39.4%) adults, respectively When the study participants were categorized, one hundred and sixty-four (68.9%) were found to belong to a scheduled tribe, eight (3.36%) belonged to a scheduled caste, eighteen (7.56%) belonged to other backward class, and forty-eight (20.16%) belonged to the general category. Among the participants, thirty-three were tested positive for a trait (AS), three had the disease (SS), and the rest were normal (AA) (Table 3). All the point-of-care results were in agreement with the HPLC (Table 4). The category-wise prevalence of sickle cell disease and traits is given in Table 3. The overall prevalence of the sickle cell gene was 15.1%, and the prevalence of the sickle cell trait and SCD was 13.86% and 1.26%, respectively. The prevalence of SCD among the ST communities was 11.34%, and it was the highest among all.

The distribution of participants screened by age included three groups: children, who make up the largest segment at 44.95% with 107 individuals; followed by adults with 94 individuals accounting for 39.49%; and adolescents, who are the smallest group with 37 individuals, representing 15.54% of the total.

The analysis confirmed that the 10% of normal samples (21 AA samples), which were also tested using HPLC, were accounted for in the existing analysis. All these samples were confirmed as true negatives, as mentioned in Table 4, perfectly aligning with the HemoType SC POCT results. This confirms that the specificity and negative predictive value reported in the table are accurate, with both parameters achieving 100%, as mentioned in Table 5. There were no false positives or false negatives among these samples, supporting the robustness of the testing methods used in this study. Thus, the sensitivity, specificity, and predictive values remain unchanged and accurately reflect the performance of the diagnostic tests as initially reported. However, it is important to consider that the sample size in this study is relatively small, which could limit the generalizability of the results. The confidence intervals for some metrics, particularly specificity and negative predictive value, are fairly wide, indicating some uncertainty in these estimates.

## 5. Discussion

Primary care must become the focal point of sickle cell disease management, with a focus on initiatives that reach a sizable segment of the population and make use of low-cost, basic technology [14]. Among the many benefits of point-of-care are (i) ease of use by local staff; (ii) quick results delivery, which allows for timely patient notification and, if needed, family counseling; and (iii) the ability to use the test in remote locations, which leads to early diagnosis and lowers mortality and morbidity through the use of therapeutic interventions, particularly anti-malarials and penicillin prophylaxis [10]. This present study was carried out in the tribal sub-plan districts. People live in a scattered manner in tribal plan districts. Primary healthcare centers have been established per 5000 people, but in the tribal sub-plan districts, they have been established per 3000 people. Therefore, the use of POC for SCD screening is more relevant in the tribal sub-plan districts. This study revealed that healthcare workers in the primary care set-up are competent in performing the POCT for the diagnosis of sickle cell disease. This study also demonstrated that screening children for SCD using point-of-care testing (POCT) through healthcare workers at the health sub-center and primary healthcare centers is feasible. The POCT has the potential to be implemented on a large scale and could be fully integrated into the National Sickle Cell Anemia Control Mission (NSCACM). Our study showed that the sensitivity and specificity of the POCT was 100%. Similarly, 100% sensitivity and specificity for SCD were found in research on point-of-care screening for SCD in low-resource settings [15]. Another study that examined the viability of integrating cutting-edge point-of-care test devices into current immunization programs in primary healthcare settings in Nigeria revealed that the HemoTypeSC test had 100% sensitivity and specificity in detecting sickle cell disease [10]. For the successful implementation of NSCACM, newborn screening (NBS) will benefit from the promotion of conducting pilot studies on the commencement of point-of-care diagnosis for SCD by healthcare workers in the primary healthcare system.

In this study, the prevalence of the SC gene was 15.12% in a population of 238. The prevalence of this present study is close to those reported by Mohanty et al. [1,16]. This prevalence in the tribes and geographical region seems stable. Promoting early screening in Rajasthan Province in India will be one of the best strategies to help prevent morbidity and mortality related to sickle cell disease. Similar observations were also made by Danho et al. [17].

Mukherjee et al. [18] screened a total of 1559 individuals (980 newborns and 579 adults) from four parts of India and analyzed them using both methods. HemoTypeSC demonstrated a sensitivity and specificity of 98.1% and 99.1% for every potential phenotype (HbAA, HbAS, and HbSS). The majority of laboratories in isolated tribal communities do not have HPLC, and it is rather costly. The current study regions are comparable to this. The conclusion was made that newborn and population screening for the HbS phenotype can be performed with the quick, point-of-care HemoTypeSC test.

In this trial, every newborn with a SCD diagnosis was enrolled right away in the SCD clinic for treatment tailored to their condition. Therefore, it is crucial to implement universal early infant screening, which enables the diagnosis and start of preventative treatments before symptoms appear. In order to increase accessibility and sustainability, this study shows that it is feasible to include early infants in the screening program as part of the regular immunization programs in primary healthcare facilities and health sub-centers. An early infant screening program for sickle cell disease (SCD) can be established by utilizing the sustainable nature of immunization as a basic healthcare component. The strategy employed in this study using PHCs and sub-centers seems to have promoted the viability of the screening program.

## 6. Conclusions

The results of this study conclude that the healthcare workers in the region are competent to perform the POCT for the diagnosis of sickle cell disease. The POCT may be introduced in the program for the diagnosis of SCD. However, integrating POCT into the NSCACM will require a thorough cost-effective analysis and implementation research. These studies will provide crucial data on the financial and practical aspects of deploying POCT across various healthcare settings. Once these analyses confirm the viability and benefits of POCT integration, we can proceed with incorporating it into the NSCACM, ensuring that it adds value efficiently and enhances patient outcomes without undue financial strain on the healthcare system.

Limitations of the study: The medical officer’s supporting staff cleaned the biomedical waste generated during the medical officer’s diagnosis. Hence, the competence in the post-analytical phase of MO was not comparable with the LTs and ANMs. The doctor did not carry out the post-analytical phase. Thirteen individuals, more than the sample size, participated in this study.

## Figures and Tables

**Figure 1 diagnostics-15-00348-f001:**
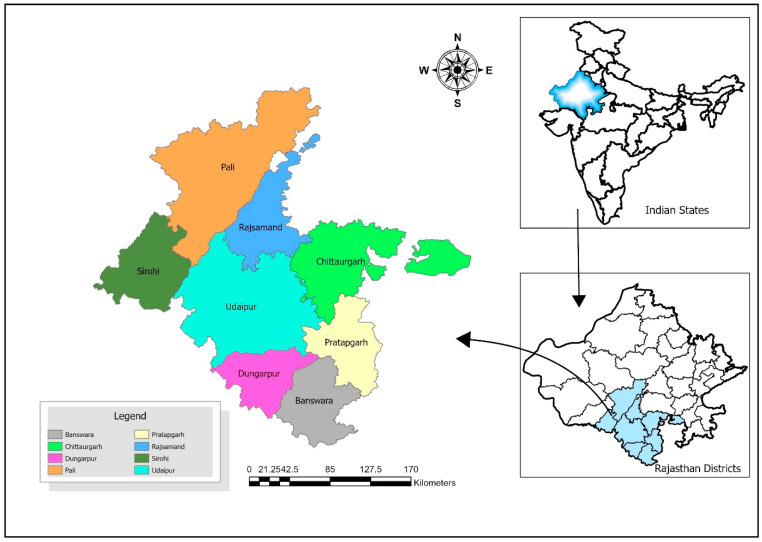
Study sites: Kotra and Jhadol blocks of Udaipur District, Rajasthan, India.

**Table 1 diagnostics-15-00348-t001:** Demographic details of the healthcare workers selected for the study.

Healthcare Workers	ANM	LT	MO	Total (*n* = 25)
Sex Distribution				
Male	0	9 (90%)	5 (100%)	14 (56%)
Female	10 (100%)	1 (10%)	0	11 (44%)
Caste/Community				
General	0	1 (10%)	1 (20%)	2 (8%)
Other Backward Classes (OBC)	1 (10%)	2 (20%)	0	3 (12%)
Scheduled Castes (SC)	2 (20%)	1 (10%)	0	3 (12%)
Scheduled Tribes (ST)	7 (70%)	6 (60%)	4 (80%)	17 (68%)
Age Groups				
18–28 Years	5 (50%)	4 (40%)	4 (80%)	13 (52%)
29–38 Years	4 (40%)	4 (40%)	1 (20%)	9 (36%)
39–48 Years	1 (10%)	2 (20%)	0	3 (12%)
Years of Experience				
<5 Years	7 (70%)	4 (40%)	4 (80%)	15 (60%)
6–10 Years	2 (20%)	3 (30%)	1 (20%)	6 (24%)
>10 Years	1 (10%)	3 (30%)	0	4 (16%)
Education Levels				
Diploma	10 (100%)	9 (90%)	0	19 (76%)
Graduate	0	0	5 (100%)	5 (20%)
Post Graduate	0	1 (10%)	0	1 (4%)

**Table 2 diagnostics-15-00348-t002:** List of checkpoints used for observation by the healthcare workers.

	Check Point	LT *n* (%)	MO *n* (%)	ANM *n* (%)	*p* Value
Pre-analytical phase	Handwash was properly conducted by the HCW	68 (97.1)	27 (96)	103 (73.6)	<0.001 *
	HCW wore gloves and PPE kit before the test	31 (44.3)	17 (60)	103 (73.6)	<0.001 *
	HCW decontaminated the working surface	66 (94.3)	27 (96)	109 (77.9)	<0.001 *
	HCW checked the weight, height, and fever of the newborn	43 (61.4)	0 (0)	114 (81.4)	<0.001 *
	Disinfectant/Spirit used on the surface for the collection of the blood sample	70 (100)	24 (84)	129 (92.1)	<0.05
	Total-1 ^#^	56 (79.4)	24 (84)	112 (79.7)	<0.001 *
Analytical phase	Properly opened and handled the kit	68 (97.1)	28 (100)	113 (80.7)	<0.001 *
	Used the accessories designated for the test	70 (100)	28 (100)	124 (88.6)	<0.001 *
	Waited for the duration for declaring the result	56 (80)	24 (84)	127 (90.7)	0.091
	Checked the internal control before concluding the test	69 (98.6)	28 (100)	138 (98.6)	0.81
	HCW followed the SOP	52 (74.3)	21 (76)	125 (89.3)	<0.05
	Total-2 ^#^	63 (90)	26 (92)	125 (89.6)	<0.001 *
Post-analytical phase	Properly reviewed the results and entered them in permanent records	66 (94.6)	24 (84)	129 (92.4)	0.366
	Decontaminated the surface after completion of the work or not	64 (91.4)	22 (80)	101 (72.1)	<0.001 *
	Managed the biomedical waste	70 (100)	28 (100)	82 (58.6)	<0.001 *
	HCW counseled the women	66 (94.3)	28 (100)	128 (91.7)	0.235
	HCW referred the infants/women (sickle cell-positive) to the MO of nearby PHC or CHC	70 (100)	28 (100)	129 (92)	<0.05
	Total-3 ^#^	67 (96.1)	26 (92.8)	114 (81.3)	<0.001 *
	Grand Total ^#^	63 (89.5)	25 (91.07)	116 (83.1)	<0.001 *

Last Column presents the significance level indicated using Chi-square statistics with * *p* < 0.001. **^#^** Kruskal–Wallis test was used to assess the statistical significance of differences between groups.

**Table 3 diagnostics-15-00348-t003:** Prevalence of sickle cell gene among different ethnic groups.

	ST, *n* (%)	SC, *n* (%)	OBC, *n* (%)	General, *n* (%)	Total, *n* (%)
AS	24 (14.63)	1 (12.5)	1 (5.55)	7 (2.94)	33 (13.86)
SS	3 (1.82)	0	0	0	3 (1.26)
AA	137 (83.53)	7 (87.5)	17 (49.5)	41 (17.22)	202 (84.87)
Total	164 (68.9%)	8 (3.36%)	18 (7.56%)	48 (20.16%)	238

**Table 4 diagnostics-15-00348-t004:** Concordance of the results of POC with HPLC.

Number of Tests Conducted	HemotypeSC, *n*	HPLC, *n*	Concordance%
AS	33	33	100%
SS	03	03	100%
AA	21	21	100%

**Table 5 diagnostics-15-00348-t005:** Diagnostic test performance metrics for sickle cell disease detection.

Metric	Value (%)	95% Confidence Interval
Sensitivity	100.00%	90.26% to 100.00%
Specificity	100.00%	83.89% to 100.00%
Positive Predictive Value	100.00%	90.26% to 100.00%
Negative Predictive Value	100.00%	83.89% to 100.00%

## Data Availability

The data will be available on request from the corresponding author. The data are not publicly available due to privacy.

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
