# Peer review of "Operational Feasibility of Point-of-Care Testing for Sickle Cell Disease in Resource-Limited Settings of Tribal Sub-Plan Region of India"

_diagnostics, 2025, doi:10.3390/diagnostics15030348_

Round 1
Reviewer 1 Report
Comments and Suggestions for Authors
The manuscript is well-organized, with a clear structure from introduction to conclusion. However, there are areas where the clarity of writing, depth of analysis, and rigor in methodology presentation can be improved. The following comments can be implemented to improve its scientific rigour and publication quality:
1. A speicific roadmap for integrating POCT into the National Sickle Cell Anemia Control Mission (NSCACM) can be suggested and included, if possible
2. Cost analysis for any process is important, so compare POCT with other available low-cost diagnostic tools to provide a comprehensive cost-benefit analysis.
3. The study does not provide data on the long-term clinical outcomes of infants diagnosed and treated based on POCT results, follow up studies can be conducted.
Author Response
1. Comments: A specific roadmap for integrating POCT into the National Sickle Cell Anemia Control Mission (NSCACM) can be suggested and included, if possible:
Response and Action Taken:
It is always very important and useful to integrate such type of POCT into the program. Added the following paragraph in the conclusion part of the manuscript. |
However, integrating POCT into the National Sickle Cell Anemia Control Mission (NSCACM) will require a thorough cost-effective analysis and implementation research. These studies will provide crucial data on the financial and practical aspects of deploying POCT across various healthcare settings. Once these analyses confirm the viability and benefits of POCT integration, we can proceed with incorporating it into the NSCACM, ensuring that it adds value efficiently and enhances patient outcomes without undue financial strain on the healthcare system.
Page No-10, Line No-243-247.
2. Comments: Cost analysis for any process is important, so compare POCT with other available low-cost diagnostic tools to provide a comprehensive cost-benefit analysis.
Response and Action Taken: The aim of the study was not to analyze the cost-benefit ratio and cost-effectiveness. However, a future study may be planned for the same.
3. Comments: The study does not provide data on the long-term clinical outcomes of infants diagnosed and treated based on POCT results, follow up studies can be conducted.
Response and Action Taken: Due to budget constraints, extensive follow-up studies are not feasible at this stage. However, once POCT is fully integrated into the National Sickle Cell Anemia Control Mission, securing additional funding for such studies could become more feasible by maintaining a registry of affected individuals.

Reviewer 2 Report
Comments and Suggestions for Authors
Abstract:
- Terms are written in full before being used as an abbreviation, e.g. ANMs, LTs, MOs, SOPs.
Introduction
- Page 2, line 34; this statement was mentioned in the introduction of reference 1. It is better to change the reference. Suggested reference: Tebbi CK. Sickle Cell Disease, a Review. hemato. 2022;3(2):341-366.
- Page 2, last paragraph; the statement have been mentioned above in lines 40-41. It is better to delete it.
- Page 3, lines 60-67; A reference needs to be added to the POCT properties. In addition, Authors need to put the properties in one paragraph rather than separated points, e.g. it is affordable and cheap , each kit costs less than 0.5 USD, the test kit does not require much training or specific skills, requires 1-5 l finger prick blood samples, with rapid results within 10 minutes. Also, most literatures mentioned that it costs less than 2 USD.
- Aim of the study is to assess the competence rather than the feasibility of health workers.
Methodology
- Page 3, The first statement is incomplete.
- Page 4, line 89; Do you mean 249,297?
- The protocol for the training of health workers listed below needs to be summarized and to be put in one paragraph.
- Page 7, a reference needs to be added about the national ANC guidelines.
- Page 7; It is very important to clarify this point; do LT have to carry 10 tests on participants to diagnose SCD? OR they have to diagnose 10 cases of SCD as written in the next page, line 131? The same is applied to MO.
- The observation checklist needs to be added to the research as a supplementary file.
Results
- Table 1 needs more clarification and major changes; the total number should be corrected, it is very important to add subheading for each category; e.g. age, sex, education ----.
- Table 1; What authors mean by general, OBC, SC, ST? These abbreviations were not mentioned before.
- Table 2; the number should be added before each percentage. The P value for the studied variables need to be added to this table also to look for statistical significance.
- It is enough to mention the age of participants in the paragraph above and there is no need for Table 3.
- Table 4 needs to deleted and the numbers of participants from different ethnic groups can be added to Table 5.
- In the Methods, Page 6, line 120, Authors mentioned that 10% of the normal samples were also run through the HPLC . These results need to be added to this table to look for sensitivity, specificity and predictive values.
Discussion
- To conclude that the POCT test is 100% sensitive requires the results of testing of the 10% of the normal samples through the HPLC to the last table in the results.
References
- References need to be written according to the journal style.

Author Response
Comments-1: Abstract: Terms need full forms before abbreviations.
Response-1: Terms have been fully spelled out before their first use and further used as abbreviations throughout the document. (Page No: 1, Line No: 13-18)
Comments-2: Introduction: Page 2, line 34; this statement was mentioned in the introduction of reference 1. It is better to change the reference. Suggested reference: Tebbi CK. Sickle Cell Disease, a Review. hemato. 2022;3(2):341-366.
Response-2: As suggested, the reference has been updated to Tebbi CK's review on Sickle Cell Disease. Ref: Tebbi CK. Sickle cell disease, a review. Hemato. 2022 May 30;3(2):341-66. (Page No: 11, Line No: 264.)
Comments-3: Introduction: Page 2, last paragraph; the statement have been mentioned above in lines 40-41. It is better to delete it.
Response-3: As suggested, the statement have been removed from the paragraph of Page 2. (Page No: 2 Line No: 53)
Comments-4:
Introduction: - Page 3, lines 60-67; A reference needs to be added to the POCT properties. In addition, Authors need to put the properties in one paragraph rather than separated points, e.g. it is affordable and cheap , each kit costs less than 0.5 USD, the test kit does not require much training or specific skills, requires 1-5 l finger prick blood samples, with rapid results within 10 minutes. Also, most literatures mentioned that it costs less than 2 USD.
Response-4: A reference has been added for the properties of POCT:
Reference: Nnodu, O., Isa, H., Nwegbu, M., Ohiaeri, C., Adegoke, S., Chianumba, R., et al. (2019). HemoTypeSC, a low-cost point-of-care testing device for sickle cell disease: Promises and challenges. Blood Cells Mol. Dis. 78, 22–28. doi:10.1016/J.BCMD.2019.01.007
Lines 60-67 have been modified as suggested:
Health departments across many Indian states are considering the POCT (lateral flow device) for diagnosing SCD due to its advantages[12]. It is affordable and cheap; each kit costs less than 0.5 USD, the test kit does not require much training or specific skills, is user-friendly with minimal training needed, and requires only a small blood sample (1-5L) from a finger prick. The POCT delivers results within 10 minutes, offering a quicker, simpler alternative to HPLC, which requires more blood and complex equipment[10]. Additionally, it is portable and does not need electricity, making it suitable for use in various settings. (Page No: 2, Line Number: 58-63)
Comments-5: Aim of the study is to assess the competence rather than the feasibility of health workers.
Response-5: The suggestions have been incorporated and modified sentence is given below-
This study will assess the competence of healthcare workers in testing infants, adults, and pregnant women for sickle cell disorder using point-of-care devices. (Page No: 2, Line No: 65)
Comments-6: Methodology: Page 3, The first statement is incomplete.
Response-6: Incomplete statement reviewed and corrected:
This study was cross-sectional in nature and was conducted to assess the competence of healthcare workers for the diagnosis of SCD through Haemotype-SC. (Page No: 2, Line No: 68-69)
Comments-7: Methodology: Page 4, line 89; Do you mean 249,297?
Response-7: Yes. (Page No: 3, Line No: 83-84)
Comments-8: Methodology: Summarize the training protocol.
Response-8: Training protocol is summarized as follows: Health workers, including ANMs, LTs, and MOs, were trained on the HemoTypeSC™ rapid test kit during a half-day workshop, which focused on identifying crucial hemoglobin variants such as HbAA, HbSS, HbSC, HbCC, HbAS, and HbAC for screening sickle cell disease and other hemoglobinopathies. The training covered the kit's components—Test Strips, Blood Sampling Devices, and Dropper Pipettes—and emphasized biohazard safety in the setup of the testing area. It was noted that additional infrastructure at the healthcare facilities like potable water, timers, lancing devices, test vials, and racks were necessary but doesn’t exist and were provided by the research team.
The POCT testing platform was established at PHC/Sub-Centre locations, where a step-by-step approach for using the HemoTypeSC was demonstrated. During practical sessions, trainees practiced blood sample collection, ensuring the white pad of the Blood Sampling Device was fully saturated. They learned the procedure steps: adding water to the Test Vial, obtaining a blood sample, swirling the sample in the vial to mix, inserting and then reading the Test Strip against a Results Chart after 10 minutes.
Before conducting the SCD testing, consent was obtained from all eligible participants. The results of the HemoTypeSC were then compared with those from HPLC testing. Participants who tested positive were subsequently referred to the nearby PHC. The training emphasized correct technique, result interpretation, and post-test procedures including the disposal of used components, all to ensure proficiency in using the HemoTypeSC™ test kit effectively. (Page No: 5, Line No: 97-110)
Comments-9: Page 7, a reference needs to be added about the national ANC guidelines.
Response-9:
A reference has been added regarding national ANC guidelines: https://nhm.gov.in/images/pdf/programmes/maternal-health/guidelines/sba_guidelines_for_skilled_attendance_at_birth.pdf (Page No: 5, Line No: 128)
Comments-10: Methodology: Page 7; It is very important to clarify this point; do LT have to carry 10 tests on participants to diagnose SCD? OR they have to diagnose 10 cases of SCD as written in the next page, line 131? The same is applied to MO.
Response-10: Corrected statement: Each LT carried out the point of care test on 10 different participants to diagnose sickle cell disease. Similarly, the medical officer conducted point of care test on 5 different participants to diagnose sickle cell disease. (Page No: 5, Line No: 132-134.)
Comments-11: Results: Table 1 needs clearer subheadings and corrected totals.
Response-11: Table 1 was revised with clear subheadings for age, sex, education, etc., and the total numbers were corrected. (Page No: 5-6 Line No: 156-157.)
Comments-12: Table 1; What authors mean by general, OBC, SC, ST? These abbreviations were not mentioned before.
Response-12: Full form of OBC (Other Backward Classes), SC (Scheduled Castes), ST (Scheduled Tribes) has been mentioned in the table. (Page No: 5-6 Line No: 156-157.)
Comments-13: Results: Add numbers and P values in Table 2 for statistical significance.
Response-13: Table 2 was updated to include numbers alongside percentages and P values added for assessing statistical significance. (Page No: 7-8 Line No: 175-176.)
Comments-14: It is enough to mention the age of participants in the paragraph above and there is no need for Table 3.
Response-14: Deleted Table 3 and added the content of the table in the paragraph.
The distribution of participants screened by age included three groups: Children, who make up the largest segment at 44.95% with 107 individuals, followed by Adults with 94 individuals accounting for 39.49%, and Adolescents, who are the smallest group with 37 individuals representing 15.54% of the total. (Page No: 8 Line No: 186-188.)
Comments-15: Results: Remove Table 4; integrate into Table 5.
Response-15: Table 4 was removed, and relevant data was integrated into Table 3.(Page No: 8 Line No: 198.)
Comments-16: In the Methods, Page 6, line 120, Authors mentioned that 10% of the normal samples were also run through the HPLC . These results need to be added to this table to look for sensitivity, specificity and predictive values.
Response-16: Added as a table 4 and 5 and also in the result section.
The analysis confirmed that the 10% of normal samples (21 AA samples) , which were also tested using HPLC, have been accounted for in the existing analysis. All these samples were confirmed as true negatives as mentioned in Table 4, perfectly aligning with the Hemotype SC POCT results. This confirms the specificity and negative predictive value reported in the table are accurate, with both parameters achieving 100% as mentioned in Table 5. There were no false positives or false negatives among these samples, supporting the robustness of the testing methods used in the study. Thus, the sensitivity, specificity, and predictive values remain unchanged and accurately reflect the performance of the diagnostic tests as initially reported. However, it is important to consider that the sample size in this study is relatively small, which could limit the generalizability of the results. The confidence intervals for some metrics, particularly specificity and negative predictive value, are fairly wide, indicating some uncertainty in these estimates. (Page No: 8 Line No: 189-191)
Comments-17: To conclude that the POCT test is 100% sensitive requires the results of testing of the 10% of the normal samples through the HPLC to the last table in the results
Response-17: This has been addressed in the above paragraph.
Comments-18: References: Follow journal style.
Response-18: All the references have been arranged as per the style of the Journal. (Page No:10-12.)

Round 2
Reviewer 2 Report
Comments and Suggestions for Authors
Dear Authors
As P value was added to Table 2, it is necessary to add statistical analysis to the Methods section.